# Assessment of the Impact of Alcohol Consumption Patterns on Heart Rate Variability by Machine Learning in Healthy Young Adults

**DOI:** 10.3390/medicina57090956

**Published:** 2021-09-11

**Authors:** Gheorghe Nicusor Pop, Ruxandra Christodorescu, Dana Emilia Velimirovici, Raluca Sosdean, Miruna Corbu, Olivia Bodea, Mihaela Valcovici, Simona Dragan

**Affiliations:** 1Department VI Cardiology, “Victor Babes” University of Medicine and Pharmacy, Timisoara 300041, Romania; pop.nicusor@umft.ro (G.N.P.); danavelimirovici@yahoo.com (D.E.V.); ralusosdean@yahoo.com (R.S.); miruna.corbu@umft.ro (M.C.); mihaeladanielacardio@gmail.com (M.V.); simona.dragan@umft.ro (S.D.); 2Institute of Cardiovascular Diseases, Timisoara 300310, Romania; olivia_maria88@yahoo.com

**Keywords:** alcohol, drinking pattern, binge drinking, heavy drinking, healthy young, HRV, heart rate variability, machine learning

## Abstract

*Background and Objectives:* Autonomic nervous system (ANS) dysfunction is present in early stages of alcohol abuse and increases the likelihood of cardiovascular events. Given the nonlinear pattern of dynamic interaction between sympathetic nervous system (SNS) and para sympathetic nervous system (PNS) and the complex relationship with lifestyle factors, machine learning (ML) algorithms are best suited for analyzing alcohol impact over heart rate variability (HRV), because they allow the analysis of complex interactions between multiple variables. This study aimed to characterize autonomic nervous system dysfunction by analysis of HRV correlated with cardiovascular risk factors in young individuals by using machine learning. *Materials and Methods:* Total of 142 young adults (28.4 ± 4.34 years) agreed to participate in the study. Alcohol intake and drinking patterns were assessed by the AUDIT (Alcohol Use Disorders Identification Test) questionnaire and the YAI (Yearly Alcohol Intake) index. A short 5-min HRV evaluation was performed. Post-hoc analysis and machine learning algorithms were used to assess the impact of alcohol intake on HRV. *Results:* Binge drinkers presented slight modification in the frequency domain. Heavy drinkers had significantly lower time-domain values: standard deviation of RR intervals (SDNN) and root mean square of the successive differences (RMSSD), compared to casual and binge drinkers. High frequency (HF) values were significantly lower in heavy drinkers (*p* = 0.002). The higher low-to-high frequency ratio (LF/HF) that we found in heavy drinkers was interpreted as parasympathetic inhibition. Gradient boosting machine learner regression showed that age and alcohol consumption had the biggest scaled impact on the analyzed HRV parameters, followed by smoking, anxiety, depression, and body mass index. Gender and physical activity had the lowest impact on HRV. *Conclusions:* In healthy young adults, high alcohol intake has a negative impact on HRV in both time and frequency-domains. In parameters like HRV, where a multitude of risk factors can influence measurements, artificial intelligence algorithms seem to be a viable alternative for correct assessment.

## 1. Introduction

Excessive alcohol consumption represents a health problem worldwide, which is associated with increased morbidity and mortality [1]. These are explained, at least partially, by the increased cardiovascular (CV) risk [2] of alcohol-dependent individuals. Alcohol has been associated with increased blood pressure, arrhythmias (mainly atrial fibrillation), hemorrhagic stroke, liver cirrhosis, injuries, and cancers of the liver, colorectum, breast, and upper digestive tract [3,4].

Given the latest WHO report [1], which showed an increased prevalence of binge drinking among young people and the harmful effects of excessive consumption, it is necessary to raise awareness of these issues among young people before any harm is done.

It is generally accepted that long-term exposure to alcohol affects various organs and systems, especially the nervous system. The increased likelihood of alcohol abusers to suffer CV events is explained, particularly, by the close relationship between the autonomic nervous system (ANS) and the heart and vessels [5]. In alcohol abuse, the ANS dysfunction is determined mostly by the inhibition of the parasympathetic nervous system (PNS) and/or the hyperactivity of the sympathetic nervous system (SNS) expressed by a sympathetic hyperarousal (tachycardia, diaphoresis, elevated blood pressure) [6]. The ANS imbalance can be estimated by analysis of heart rate variability (HRV), determined by means of Holter ECG monitoring. On the other hand, some authors suggest that the correlation between a change in HRV and altered morbidity and mortality is substantially attributable to the concurrent change in HR [7,8]. When dealing with patients with comorbidities, HRV cannot be used in any simple way to assess autonomic nerve activity to the heart [9].

HRV analysis, in time or frequency domain, is a reliable, noninvasive method for the assessment of autonomic functions, by several parameters currently used to characterize the physiological spontaneous fluctuations in heart rate (HR) and normal R-R intervals [10]. This method is used in physiological models and various pathological states, for the assessment of cardiovascular risk [11,12,13]. Certain parameters as the root mean square of the successive differences (RMSSD) and high-frequency power (HF) characterize mainly the PNS function [14].

Most of the studies regarding alcohol effects on HRV that we found scanning relevant literature could be divided into two categories. In the first category were small-scale studies focused on HRV changes after acute alcohol ingestion, including only a low number of participants (between 8 and 36) [15,16,17]. The second category approached HRV alterations in chronic alcohol intake. The studies in this category usually compared participants with alcohol dependence versus control, and included mostly men aged over 36 years [18,19,20].

The similarities and differences between statistics and machine learning are a topic that generates plenty of discussions. It is generally accepted that statistics focus on hypothesis testing and inference while ML focuses on data fitting and predictive accuracy [21]. In complex situations like HRV, where SNS and PNS dynamic interaction follow a non-linear pattern [7,22], the ML algorithms can identify patterns and trends that might not be apparent to a human used to ordinary statistics. The benefits of ML include superior performance and accuracy in problems where the relationships between factors are complex. Another benefit of ML stands in its validation process, which helps better use of data, and gives much more information about the algorithm performance.

Our study addressed a group of young healthy individuals 18 to 35 years old, both males and females, who admitted alcohol consumption. It was aimed to characterize ANS dysfunction by analysis of heart rate variability (HRV) in different patterns of alcohol intake and also to correlate HRV with cardiovascular risk factors using machine learning, in an attempt to evaluate the complex interaction between these variables.

## 2. Materials and Methods

### 2.1. Participants

Our study was conducted between 15 January and 15 March 2020, in the outpatient service of a Cardiovascular Prevention and Rehabilitation Clinic. Our evaluations were precociously terminated due to the COVID-19 pandemic outbreak. It followed a prospective, cross-sectional design and targeted a group of healthy young adults, without any history of chronic pathologies or previous Anonymous Alcoholics records. They were recruited from subjects attending the hospital outpatient department for minor complaints or a routine exam. After the anamnesis and a detailed clinical exam, we excluded all subjects with a history of previous or present chronic diseases, drug use, and also performance athletes. To avoid bias, we also excluded subjects who were abstinent from alcohol. Total of 142 young adults agreed to participate in our study and all of them signed a written informed consent form. There were 96 (67.6%) men and 46 (32.3%) women, aged between 18 and 35 years (mean age 28.44 ± 4.34). All of them admitted alcohol consumption but considered that their intake was within normal limits.

### 2.2. Instruments

We assessed alcohol consumption and drinking behavior using the validated Romanian version of the AUDIT questionnaire (Alcohol Use Disorder Identification Test), which is a 10-item screening tool developed by the World Health Organization (WHO) [23]. Participants were questioned about the preferred beverages they usually consume (beer, wine, spirits), weekly alcohol intake, and the age when they started drinking. The AUDIT questionnaire comprises 3 sub-scales: consumption score (3 questions, maximum possible score = 12; 6 or higher may indicate a risk of alcohol-related harm), dependence score (3 questions, maximum possible score = 12; 4 or higher suggests alcohol dependence), and self-perceived alcohol-related personal problems (the last 4 questions of the questionnaire). The last section addresses feelings of guilt or remorse after drinking, having trouble remembering things after drinking, concerns of family or friends about their drinking, and injury inflicted upon someone while being under the influence of alcohol. Scoring any point in this last section requires further investigation to determine whether the problem is of current concern and requires intervention.

The quantity of alcohol consumed was expressed in alcohol units, which measure the estimated pure alcohol content of a drink. In Romania, by consensus, one unit of alcohol is equivalent to 12 g of pure alcohol [24]. The number of alcohol units in each serving of beverage depends on its size and concentration. In our study, 330 mL of beer, or 125 mL of wine, or 40 mL of spirits represented 1 unit of alcohol.

To further assess consumption, we defined the Yearly Alcohol Intake (YAI-index), by using a simple formula that relates to weekly alcohol intake (units) multiplied by the number of years of consumption.

Based on the results of the AUDIT questionnaire and the YAI-index we defined three groups:Casual drinkers: who usually drink only occasionally low quantities of alcohol (e.g., a toast at a celebration), corresponding to a consumption of 4 units or less in the past week. These subjects scored ≤3 points on the consumption sub-scale of the AUDIT questionnaire.Binge drinkers: who consume at least 6 units of alcohol on repeated occasions at least once per month. These subjects scored >3 points on the consumption sub-scale of the AUDIT questionnaire.Heavy drinkers: who consume weekly at least 16 units in men and 10 units in women [25].

Smoking status was assessed using short questions about smoking habits. Two groups were identified: current or former smokers and non-smokers. Pack-years index was calculated for standard manufactured cigarettes, by multiplying the number of packs smoked per day by the number of years the person has smoked [26].

For the assessment of anxiety and depression, we used the HADS questionnaire [27], which is a 14-item self-rating assessment tool that consists of a 7-item subscale for both depression and anxiety. A compiled score of 7 or greater in the subscale indicates symptoms of mental disorder and a score of 10 or greater indicates clinically significant anxiety or depression.

Participants were asked whether they considered themselves sedentary (reporting that during their spare time they mostly read, watch TV, and spend time in ways that do not imply physical activities) or physically active (engaged in some kind of moderately strenuous activity, e.g., walking at least 4 h/week or >30 min/day).

### 2.3. HRV Methods

The participants were instructed to take the HRV test after at least 24 h abstinence from alcohol and 12 h from smoking. To avoid potential confounding effects of circadian rhythm, all assessments were performed in the morning between 8:00 and 10:00 AM in a quiet, temperature-controlled room (24 °C), in dorsal decubitus.

To assess HRV, after a resting period of 5 min in supine position, we performed a short 5-min monitoring, with a Polar H10 device (Polar Electro Oy, Kempele, Finland) at 5000 Hz. Respiration rate does not markedly influence HRV during resting state recordings [28]; therefore, participants were instructed to breathe normally for the duration of the recording. All measurements were performed in accordance with manufacturer user manuals.

Raw data were exported and inspected for errors. After a valid recording, the data was examined with Kubios HRV 3.4.3 software (Kubios Oy, Kuopio, Finland) and the samples were filtered with the low automatic filter, and visually inspected for artifacts. A summary of HRV parameters calculated by the Kubios HRV software is represented in Table 1.

### 2.4. Statistical Analysis

Continuous variables are presented as mean and standard deviation (SD) or median and interquartile range [IQR], and categorical variables are presented as frequency and percentages. We performed descriptive and inferential statistical analysis to summarize the characteristics of the study population. The results of the Shapiro–Wilk normality test showed a non-Gaussian distribution; therefore, we continued to use non-parametric tests. To assess the differences of HRV parameters between genders we employed the Mann–Whitney U test. To evaluate general and HRV characteristics between casual, binge, and heavy drinking groups we employed the Kruskal–Wallis test, followed by post-hoc analysis with the Mann–Whitney U test with Bonferroni correction for pairwise comparison. To evaluate the proportion of various categorical variables in the groups, we applied the Chi-squared test (χ2). For the correlation analysis between various HRV parameters and alcohol intake we employed the Spearman rank test.

A *p*-value less than 0.05 was considered statistically significant. Data analysis was performed using IBM SPSS Statistics version 26 (IBM Corp, Armonk, NY, USA).

### 2.5. Machine Learning Method

To answer our primary research question on the impact of chronic alcohol consumption on various HRV parameters we employed six different machine learning (ML) algorithms: Random Forest, XGBoost Tree, Gradient Boosting, Support Vector Machine (SVM), Neural Net–Multi-Layer Perceptron (MLP), and Linear-AS. Based on the performance of these algorithms, we choose the method that showed the highest performance for further analysis (Table 5).

Age, gender, body mass index, alcohol (YAI index), smoking (Pack-Year index), physical activity (less or more than 4 h/week), anxiety, and depression (HADS questionnaire) were the classifiers used in the machine learner models.

The training was performed on a randomly selected partition consisting of 80% of our data, while the testing was performed on the remaining 20%. Training and testing of the classifiers were done by a repeated ten-fold cross-validation method. To avoid biased prediction, we averaged model performance metrics across test folds.

We made six different models of Gradient Boosting Regression where the target factors were RMSSD, SDNN, PNN50, LF, HF, and LF/HF ratio respectively. Scaled values of the classifiers’ importance are reported in Table 6. For the ML analysis we used the AutoML and H2O Gradient Boosting Machine Learner modules for KNIME Analytics Platform 4.3.1 (KNIME AG, Zurich, Switzerland).

## 3. Results

### 3.1. General Characteristics

One hundred forty-two participants, aged between 18 and 35 years, mean age 28.44 ± 4.34 years were included in this study. There were 96 men (67.6%) and 46 women (32.4%) and 70.4% of them lived in urban areas. The preferred beverage in the study population was beer (56%), followed by wine (25%), and spirits (19%).

The general characteristics of the study population are presented in Table 2.

While there were no significant differences between gender representations in the three groups, we observed that the male gender was more predominant in binge and heavy drinking groups. A significant age difference was found between the three groups, the heavy drinking group having the lowest median age.

Although most of the participants in the study came from urban areas, the distribution in the three groups was homogeneous, with no statistically significant differences. In terms of BMI, there were also no significant differences between groups. Regarding physical activity, binge drinkers were the most active, while casual drinkers and heavy drinkers reported lower physical activity.

Casual drinkers and heavy drinkers had higher anxiety scores than binge drinkers, but there were no significant differences in depression levels. Those who practice binge drinking and heavy drinking started drinking at an early age. In the heavy drinkers group we found a higher smoking incidence.

### 3.2. HRV Time-Domain Alterations

Time-domain values are presented in Table 3. There was a significant difference in SDNN between genders, indicating that females had lower SDNN values than males (34.9 vs. 46.6, *p* = 0.036, Mann–Whitney U test). The RMSSD difference between genders was not significant (39.4 vs. 37.7, *p* = 0.716, Mann–Whitney U test). Further gender comparison stratified by drinking patterns showed lower RMSSD values in women in the heavy drinking group (24 vs. 34.1, *p* < 0.001, Mann–Whitney U test), but with no statistical differences in casual drinkers and binge drinkers.

Binge drinkers had a lower minimum and mean heart rate compared with other groups, but without significant differences. Heavy drinkers had the highest maximum heart rate without significant differences compared to the two other groups.

Despite similar heart rates, the heavy drinkers had statistically significant lower SDNN, RMSSD, NN50, and pNN50 values compared to casual and binge drinkers. Binge drinkers had the lowest RRTI and TINN, but the difference between groups was not statistically significant.

Post-hoc analysis showed statistically significant differences in RMSSD and SDNN in heavy drinkers versus casual and binge drinkers. There were no significant differences between casual and binge drinkers in RMSSD and SDNN (Figure 1).

The Spearman test showed a significantly low negative correlation of RMSSD with the AUDIT score (r = −0.339, *p* < 0.001) and a moderately negative correlation with the YAI-index (r = −0.438, *p* < 0.001). SDNN presented a low negative correlation with the AUDIT score (r = −0.374, *p* = 0.001) and a moderately low correlation with the YAI-index (r = −0.483, *p* < 0.001).

### 3.3. HRV Frequency-Domain Alterations

The Frequency-Domain analysis results are presented in Table 4.

There were no statistically significant differences in VLF, LF, and LF/HF ratios between groups. In the HF spectrum, we found statistically significant differences between drinking patterns, heavy drinkers having the lowest HF. We found statistically significant differences of total power between groups, binge drinkers having the highest value.

The post-hoc analysis using the Mann–Whitney U test and *p*-values adjusted with Bonferroni correction for pairwise comparison revealed that heavy drinkers had lower values in the HF spectrum when compared to casual drinkers (*p* = 0.032) and binge drinkers (*p* = 0.029) while there was no statistically significant difference between casual and binge drinkers (*p* = 0.824).

Gender comparison indicated that females had lower LF (56.8 vs. 63.8), higher HF (42.5 vs. 36.9), and lower LF/HF ratio (1.32 vs. 1.76) versus males, but with no statistically significant differences (Mann–Whitney U test).

### 3.4. Machine Learning Algorithms

We employed six different machine learning algorithms to evaluate HRV alteration in the present study. The performance of these algorithms is presented in Table 5. The most performant algorithm was chosen for further deep analysis.

As the Gradient Boosting algorithm showed the best performance, we chose this method to assess the impact of alcohol intake on HRV correlated to other risk factors like age, gender, BMI, anxiety, depression, physical activity, and smoking.

In Table 6 we report the scaled impact of each risk factor.

Age and alcohol intake had the biggest impact on the analyzed HRV parameters, followed by smoking, anxiety, depression, and BMI. Gender and physical activity had the lowest impact on HRV.

## 4. Discussion

Chronic, excessive alcohol consumption represents the major cause of alcoholic cardiomyopathy which is associated with congestive heart failure, arrhythmias, and sudden cardiac death [30,31]. In 2018 WHO reported a worldwide decrease of binge drinking defined as 60 or more grams of pure alcohol on at least one occasion at least once per month. However, prevalence rates among drinkers of 15–24 years were higher than in the total population. Young people of 15–24 years often drink in binge drinking sessions [1].

A weekly pattern of alcohol consumption was observed in a study that included 496 participants, binge drinkers seem to have a sharp increase in consumption on weekends, while heavy drinkers showed a linear increase from Monday toward Sunday [32]. With this rise of alcohol consumption among young people and the risk that comes along, we wanted to assess the impact of alcohol intake patterns on the cardiovascular system.

### 4.1. Assessment of Alcohol Intake

Three different lifetime drinking measures are present in the literature: Lifetime Drinking History (LDH), Concordia Lifetime Drinking Questionnaire (CLDQ), and Cognitive Lifetime Drinking History (CLDH) [33,34,35]. The two latter questionnaires have seen limited use and have each been evaluated in one study. The LDH questionnaire uses a floating time interval to collect data and the attention is focused upon the frequency, variability in consumption, and types of beverages. This questionnaire has advantages when longer assessment intervals are needed, but the structure is complex and takes 20–30 min to complete and 5–10 min to score.

Since the LDH questionnaire is a complex and time-consuming method to be applied in outpatient services and alcohol consumption follows a weekly pattern cycle, we preferred the Yearly Alcohol Intake (YAI) index. The YAI-index formula is simple: weekly alcohol intake (units) multiplied by the number of years of consumption. We are aware that the weekly intake may vary over time, the person may lie or may underestimate the consumption. Alcohol consumption was underestimated by approximately 12% by the questionnaire when compared with a weekly journal of alcohol intake [36]. Overall, the simple formula of YAI-index is easy to apply in an outpatient service and has a strong correlation with HRV alteration.

### 4.2. Assessment of HRV

The idea of ANS balance or imbalance has been used since the first HRV studies in the literature [37,38]. If the PNS and SNS work on a principle of balance, it means that when one increases the other will decrease and vice-versa, although some authors disagree with this framing [39,40]. In contradiction to the balance concept, evidence has been provided that descendant influence from the neural system can trigger different changes in the PNS and SNS, whether reciprocal, independent, or even co-active [41,42]. As the human heart tonus is sympathetically engaged, the parasympathetic innervation of the heart acts as a brake.

The reliability and validity of Polar monitors to measure R-R intervals have been confirmed against electrocardiogram [43]. The Kubios HRV analysis software is also validated and used at roughly 1200 universities in 128 countries [29].

### 4.3. Effects of Cardiovascular Risk Factors on HRV

#### 4.3.1. Age and Gender

Aging is associated with decline of HRV values in healthy subjects. Aging influences both time-domain and frequency-domain parameters [44,45,46]. In our study, machine learning regression also showed that age had the biggest impact on HRV, both in time and frequency domains. As for gender, the vast majority of studies in the literature reveal that females have higher values of parasympathetic autonomic functions compared to males [44,46,47]. In our study the results were similar to the literature, the females had higher HF values, lower LF and LF/HF ratio values compared to males, which indicate a higher parasympathetic activation.

#### 4.3.2. Smoking and Body Mass Index

The CHRIS study [48] evaluated HRV among 4751 adults and showed that current smokers have lower HRV values compared to non-smokers. Heavier smoking intensity provided evidence to gradually reduce HRV values, both in time and frequency domains. In our study, the machine learning algorithm showed that smoking was the third factor of importance affecting HRV, after age and alcohol intake.

Koenig et al. [49] in a study on fifty-nine healthy adults showed that sympatho-vagal balance is related to BMI in non-obese subjects, higher BMI values being associated with parasympathetic inhibition. In our study, the majority of the participants had a normal weight. The machine learning regression showed that BMI was the sixth factor of importance affecting the HRV, with a total scaled impact of 0.3.

#### 4.3.3. Anxiety and Depression

Several findings evidenced a reduced HRV and cardiovascular diseases association, as expressed by lower values in the time-domain, in patients that are suffering from depression compared to healthy controls [50,51,52,53]. A meta-analysis based on 36 articles [54] showed that anxiety disorders are associated with lower HRV values. In our study, while there was no difference in depression, the anxiety levels were lower in the binge drinkers’ group. None of the participants had been diagnosed with anxiety/depression disorder and none of them took any medication. In our study anxiety and depression levels were similar in the three groups and apparently did not seem to influence HRV, but when we employed machine learning algorithms, we observed that anxiety and depression had a scaled impact of 0.36 and 0.34 respectively over HRV, especially on LF and LF/HF ratio, suggesting SNS activation.

### 4.4. Effects of Drinking Patterns on HRV

#### 4.4.1. Casual Drinking

Casual drinkers in the present study had similar time-domain values to the ones detected by Nunan et al. [55] in a meta-analysis of 30 studies about short-term HRV in healthy adults. The SDNN and RMSSD in our study versus the Nunan reports were 48.5 vs. 51 ms and 44.4 vs. 42 ms, respectively. In the Nunan review, the frequency-domain analysis included articles with two methods of calculation: auto-regressive methods and fast Fourier transformation (FFT). There were large discrepancies between values by using these two different methods, so we will only compare our results with those of 12 studies using the same method, FFT. In the frequency-domain our findings were similar to Nunan reports, 40.1 vs. 40 n.u. for HF and 1.53 vs. 1.7 for LF/HF ratio; only LF values were higher 59.9 vs. 47 n.u. All these similarities lead us to the conclusion that occasional low dose alcohol consumption has no negative effects on HRV.

#### 4.4.2. Binge Drinking

Binge drinkers in the present study had similar time-domain values compared to the Nunan meta-analysis. The SDNN and RMSSD in our study versus Nunan reports were 46.8 vs. 51 ms and 42 vs. 42 ms, respectively. In the frequency-domain binge drinkers had a higher LF 56.7 vs. 47 n.u., lower LF/HF ratio 1.31 vs. 1.7, and similar HF 43.2 vs. 40 n.u.. Even if the time-domain values were similar, we observe slight modification in the frequency-domain, which suggests that binge drinking has negative effects on HRV.

#### 4.4.3. Heavy Drinking

Heavy drinkers in the present study had significantly lower time-domain values than those reported in the Nunan meta-analysis. The SDNN and RMSSD in our study versus Nunan reports were 35.5 vs. 51 ms and 31.9 vs. 42 ms, respectively. In the frequency-domain heavy drinkers had lower HF 32.2 vs. 40 n.u., higher LF 65.6 vs. 47 n.u. and LF/HF ratio 1.97 vs. 1.7. Knowing that HRV decreases with age, even though HD were the youngest in our study, they obtained the worst HRV scores. This shows the huge negative impact that high alcohol consumption has, regardless of age.

In the available literature, only three studies evaluated resting HRV in active drinkers with alcohol use disorder and all compared them with controls. Individuals with alcohol use disorder exhibited lower resting HF [20,56] or lower time-series HRV [19]. Studies that evaluated HRV in patients without alcohol use disorder, reported moderate or heavy drinking. These studies are inconsistent, some of them report increased HRV among drinkers when compared with controls, others find that decreased HRV is associated with higher alcohol intake [57]. There could be numerous reasons for discrepant findings. One of them is that drinking habits vary considerably among participants and no clear data on the duration of drinking are available. The study of Kupari et al. from 1993 which comprised a wide range of drinking habits found that lower HF was associated with greater alcohol intake [58].

Numerous studies have analyzed both sympathetic and parasympathetic influences for explaining LF oscillation [59,60]. Some studies indicate that increased LF/HF ratio may be an adequate reflection of sympathetic activity [61]. In our study if we look at the LF and HF values expressed in ms^2^, we observe that between casual drinkers and heavy drinkers there is just a small difference regarding LF, while HF values are significantly lower in heavy drinkers. This means that the higher LF/HF ratio that we found in heavy drinkers is falsely interpreted as increased sympathetic activity, and rather is parasympathetic inhibition.

### 4.5. Benefits of Machine Learning Algorithms

The use of digital programs in cardiology started with software that gave interpretation to the electrocardiogram (ECG). Willems et al. [62] reported in 1991 the poor diagnostic performance of nine ECG computer software. In the past decade the machine learning algorithms started to be used more and more in cardiology. As we know, HRV is influenced by many factors and this makes the machine learning algorithms the optimum solution for analyzing such a complex phenomenon. In 2020, Agliari et al. [63] used machine learning algorithms for detecting atrial fibrillation and congestive heart failure based on 24-h Holter ECGs. They reported 85% successful identification from a 2829 sample recordings. Another study published in 2019 by Chiew et al. [64] showed that their Gradient Boosting model was superior to traditional risk prediction scores.

The benefits of ML include superior performance and accuracy in problems where the relationships between factors are complex. Another benefit of ML stands in its validation process, which helps better use of data, and gives much more information about the algorithm performance. In the present study, we used six different machine learning algorithms and a ten-fold cross-validation method in order to find the model with the highest performance. Statistics draws population inferences from a sample, and machine learning finds generalizable predictive patterns. Statistics and ML are complementary in pointing us to biologically meaningful conclusions [21,65].

In a recent paper, Johnson et al. presented in detail a guide for clinicians on relevant aspects of artificial intelligence and machine learning and applications of these methods selected from reviews referring to cardiology, and also identified how both cardiovascular medicine and general medicine could incorporate artificial intelligence in the future to enable precision cardiology and improve patient outcomes [66].

Doctors make decisions based on patient data, and cardiologists tend to have access to more data than other specialties. Although the adoption of machine learning in the daily practice of cardiologists is quite limited, sooner or later these artificial intelligence algorithms will become essential, as has happened with the rapid adoption of automated algorithms for computer vision in radiology. Machine learning may facilitate the optimal development of patient-specific models for improving diagnosis, intervention, and outcome in cardiovascular medicine [66,67].

To our knowledge the present study is the first that tries to assess alcohol impact on HRV by using machine learning.

Hillebrand et al. in a meta-analysis that included 21,988 participants showed that low HRV is associated with a 32–45% increased risk of a first cardiovascular event in a population without known cardiovascular diseases. It, therefore, supports the hypothesis that cardiac autonomic dysfunction is associated with a higher risk of cardiovascular risk [68]. Tsuji et al., in The Framingham Heart Study, based on 2501 subjects, with age starting from 26 years, stated: “A one-standard deviation decrement in the standard deviation of total normal RR intervals (natural log transformed) was associated with a hazard ratio of 1.47 for new cardiac events (95% confidence interval of 1.16 to 1.86)” [69].

In the present study, we showed that consuming large quantities of alcohol is associated with lower HRV scores, even in healthy young individuals.

Current literature tackling HRV in alcohol consumption does not offer consistent findings, the biggest problem lying in the methodology. Many results could be influenced by the lack of supplementary information about other factors that could influence HRV. Smoking status, alcohol intake history, cardiovascular fitness, and associated health conditions can affect HRV but these factors are not always considered in research papers. Moreover, there are no clear boundaries or cutoffs between normal and altered HRV values. The collection method for HRV measurement has evolved in the last years and this can offer stability to the results.

Many studies in the literature have small samples, focus on male gender, and usually compare participants with alcohol use disorder to control. The present study has a significantly larger number of participants compared to the rest of the research available about alcohol impact on HRV. This study focuses on healthy young adults without known acute or chronic diseases, without any medications, 18–35 years old, both male and female, and also presents details about factors that are known to influence HRV such as smoking, anxiety, depression, BMI, and physical activity.

### 4.6. Study Limitations

As in other studies that rely upon self-reports, and also because some participants may be ashamed of their high alcohol consumption, estimation of overall alcohol intake may be subject to bias. Another limitation is represented by the fact that the cross-sectional study design assesses simultaneously the exposure and the outcome, losing the temporal relationship. A third limitation is that we used 5-min short-term HRV recordings while a 24-h recording could bring more information about overall HRV, but this was not possible in our outpatient service. Several studies proved that a 5-min HRV recording is stable and can be applied for screening, having a strong correlation with a 24-h recording [70,71,72].

Despite of the limitations mentioned above, this was a cross-sectional study and we believe that our strict inclusion criteria and careful examination have minimized the bias.

## 5. Conclusions

The results presented in this study show clearly that high alcohol intake is associated with parasympathetic inhibition, rather than sympathetic activation. In healthy young adults, high alcohol intake had a negative impact on HRV both in the time-domain and frequency-domain.

The YAI-index helped us estimate the total amount of alcohol consumption and it can be applied with ease in the outpatient service. A good correlation with HRV alterations caused by alcohol intake was observed.

Classical data analysis provides great information in medicine. In parameters like HRV, where a multitude of risk factors can influence measurements, artificial intelligence algorithms seem to be a viable alternative for correct assessment.

In the present study, machine learning models helped us assess the impact of alcohol consumption and life style on HRV in a population of young healthy adults. In the future, such algorithms could be part of more accurate cardiovascular disease risk prediction models.

## Figures and Tables

**Figure 1 medicina-57-00956-f001:**
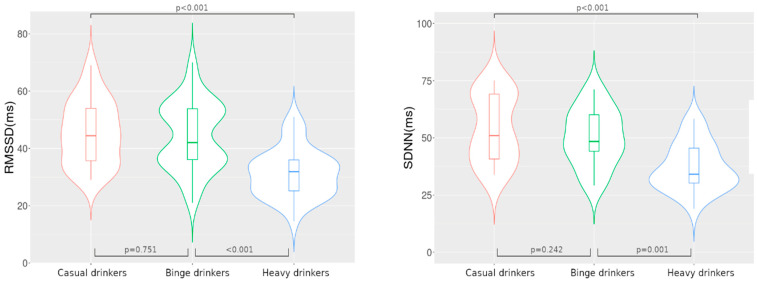
The boxplot inside violin represents the median and interquartile range. RMSSD (root mean square of the successive differences) and SDNN (standard deviation of RR intervals) violin plots of drinking patterns (casual = 45, binge = 62, heavy = 35). Post-hoc analysis using Whitney–Mann U tests. The *p*-value was adjusted with Bonferroni correction for pairwise comparison.

**Table 1 medicina-57-00956-t001:** Summary of HRV parameters calculated by Kubios HRV software [29].

Parameter	Units	Description
Time Domain		
RR	ms	The mean of RR intervals
SDNN	ms	Standard deviation of RR intervals
HR	/min	The mean heart rate
STD HR	/min	Standard deviation of instantaneous heart rate values
RMSSD	ms	Square root of the mean squared differences between successive RR intervals
NN50	count	Number of successive RR interval pairs that differ more than 50 ms
pNN50	%	NN50 divided by the total number of RR intervals
RRTI	−	The integral of the RR interval histogram divided by the height of the histogram (triangular index)
TINN	ms	Baseline width of the RR interval histogram
Frequency Domain		
VLF, LF, and HF peaks	Hz	Peak frequencies for VLF, LF and HF bands
VLF, LF, and HF powers	ms^2^	Absolute powers of VLF, LF and HF bands
VLF, LF, and HF powers	%	Relative powers of VLF, LF and HF bands
		VLF [%] = VLF [ms^2^]/total power [ms^2^] × 100%
		LF [%] = LF [ms^2^]/total power [ms^2^] × 100%
		HF [%] = HF [ms^2^]/total power [ms^2^] × 100%
LF and HF powers	n.u.	Powers of LF and HF bands in normalized units
		LF [n.u.] = LF [ms^2^]/(total power [ms^2^] − VLF [ms^2^])
		HF [n.u.] = HF [ms^2^]/(total power [ms^2^] − VLF [ms^2^])
LF/HF	−	Ratio between LF and HF band powers
Total power	ms^2^	Total spectral power

This table was adapted from [29] with the permission of Elsevier Ireland Ltd. (Dublin, Ireland), 2021. VLF = very low frequency, 0.00–0.04 Hz; LF = low frequency, 0.04–0.15 Hz; HF = high frequency, 0.15–0.40 Hz

**Table 2 medicina-57-00956-t002:** General characteristics of the study population (*n* = 142).

	Casual Drinkers *n* = 45	Binge Drinkers *n* = 62	Heavy Drinkers *n* = 35	*p*
Age (years)	28 [27–32]	30 [27–33]	27 [24–29]	0.001
Male/Female gender	24/21	46/16	26/9	0.057
Urban area	36 (80%)	40 (64.5%)	24 (68.6%)	0.215
BMI (kg/m^2^)	26.5 [21.5–30]	25 [23.5–28.5]	25 [23–28.5]	0.920
Physical activity > 4 h/week	9 (20%)	32 (51.6%)	10 (28.6%)	0.002
HADS questionnaire				
Anxiety	6 [4–9]	4 [2–7]	6 [4–8]	0.009
Depression	3 [1–4]	3 [1.75–6]	2 [1–6]	0.740
Alcohol consumption				
AUDIT questionnaire	3 [1–5]	6 [5–8]	13 [10–15]	<0.001
Weekly intake (units)	2 [0–4.5]	7 [5–9]	19 [16–21]	<0.001
Drinking start age	20 [17.75–25.5]	18 [17–21]	18 [16–20]	0.016
YAI index	19 [0–38.5]	72 [31.5–91.75]	151 [105–198]	<0.001
Alcohol type (%)				
Beer	80 [70–83.75]	70 [20–80]	50 [20–70]	0.002
Wine	10 [0–20]	10 [5–50]	30 [15–40]	0.001
Distilled Drinks	10 [8.75–15]	15 [8–30]	10 [0–35]	0.311
Smoking				
Incidence	23 (51.1%)	23 (37.1%)	32 (91.4%)	<0.001
Pack Year index	4 [2.75–11.75]	10 [4–18]	8 [3–14.25]	0.133

Median [IQR], Kruskal-Wallis Test; count (frequencies), Chi-square test.

**Table 3 medicina-57-00956-t003:** HRV–time-domain analysis (*n* = 142).

	Casual Drinkers *n* = 45	Binge Drinkers *n* = 62	Heavy Drinkers *n* = 35	*p*
RR (ms)	825 [767–852]	856 [758–940]	771 [705–883]	0.160
Min HR (/min)	68 [57–77]	61.5 [56.5–65]	65.9 [56.5–73.3]	0.127
Max HR (/min)	87 [81.6–99.3]	85 [73.7–95.7]	92 [74.6–99.4]	0.313
Mean HR (/min)	73.5 [67.8–82]	68 [65–78.8]	75.7 [64–82.7]	0.228
SDNN (ms)	48.5 [40.7–69.8]	46.8 [39.5–59.6]	35.5 [31–47.9]	<0.001 *
RMSSD (ms)	44.4 [35.1–54.5]	42 [35.9–54]	31.9 [24.6–36.1]	<0.001 *
lnRMMSD	3.79 [3.55–3.99]	3.73 [3.58–3.98]	3.46 [3.20–3.58]	<0.001 *
NN50 (count)	79.8 [44.7–98.1]]	53.3 [8.5–97.5]	39.9 [19.3–48.5]	0.049 *
pNN50 (%)	15.6 [12.2–28.6]	15.7 [11.5–32.1]	9.5 [7.8–26]	0.043 *
RRTI	11.1 [7.4–14.9]	9.7 [6.5–12.1]	10.1 [6.7–15.3]	0.424
TINN (ms)	285.2 [183–331.6]	215.1 [151.7–322.5]	249 [157–292.4]	0.248

Median [IQR], Kruskal–Wallis test; lnRMMSD-natural logarithm of RMSSD; * significance threshold value reached.

**Table 4 medicina-57-00956-t004:** HRV–frequency-domain analysis (*n* = 142).

	Casual Drinkers *n* = 45	Binge Drinkers *n* = 62	Heavy Drinkers *n* = 35	*p*
VLF peak (Hz)	0.037 [0.029–0.040]	0.037 [0.029–0.039]	0.037 [0.029–0.040]	0.706
VLF (ms^2^)	72 [29.26–132.66]	111 [53.29–173.89]	125 [49.2–401.9]	0.096
LF peak (Hz)	0.101 [0.080–0.106]	0.087 [0.067–0.114]	0.097 [0.078–0.121]	0.598
LF (ms^2^)	715.1 [437.6–1779]	1226 [954.7–1621.6]	815 [299.8–2878.7]	0.284
LF (n.u.)	59.9 [55.2–72.3]	56.7 [50.8–69.8]	65.6 [55.1–75.5]	0.180
HF peak (Hz)	0.219 [0.193–0.248]	0.212 [0.177–0.279]	0.163 [0.153–0.289]	0.042 *
HF (ms^2^)	726.1 [369.5–1015]	864.9 [674–1290.8]	392.9 [206.6–519]	0.002 *
HF (n.u.)	40.1 [28.8–43.7]	43.2 [30.8–49.1]	32.2 [25.8–42.8]	0.048 *
LF/HF ratio	1.53 [1.27–2.71]	1.31 [1.03–2.26]	1.97 [1.27–2.89]	0.165
Total power (ms)	1042 [982–3068]	2329 [1781–3173]	1064 [932–1692]	0.002 *

Median [IQR], Kruskal-Wallis test; * significance threshold value reached.

**Table 5 medicina-57-00956-t005:** Machine learning algorithms’ performance.

Algorithm	Performance *
Gradient Boosting	0.885
Neural Net (MLP)	0.877
XGBoost Tree	0.876
Random Forrest	0.812
Support-Vector Machine	0.801
Linear-AS	0.737

* Averaged model performance metrics across test folds.

**Table 6 medicina-57-00956-t006:** Gradient boosting machine learner regression (*n* = 142).

Variables	Time Domain			Frequency Domain			Total Scaled Impact
	RMSSD	SDNN	pNN50	LF	HF	LF/HF Ratio	
Age	0.631	1	1	1	1	1	1
Alcohol ^a^	1	0.741	0.504	0.451	0.246	0.767	0.66
Smoking ^b^	0.263	0.217	0.52	0.471	0.188	0.461	0.38
Anxiety ^c^	0.091	0.201	0.307	0.294	0.114	0.997	0.36
Depression ^c^	0.151	0.245	0.137	0.288	0.204	0.909	0.34
BMI	0.335	0.223	0.348	0.148	0.213	0.436	0.30
Gender	0.143	0.079	0.024	0.081	0.057	0.029	0.07
Physical activity	0.144	0.071	0.053	0.042	0.033	0.014	0.06

^a^ Yearly alcohol intake index; ^b^ Pack-Year index; ^c^ HAD questionnaire.

## Data Availability

The data that support the findings of this study are available from the corresponding author, upon reasonable request.

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
