# Peer review of "Assessment of the Impact of Alcohol Consumption Patterns on Heart Rate Variability by Machine Learning in Healthy Young Adults"

_medicina, 2021, doi:10.3390/medicina57090956_

Round 1
Reviewer 1 Report
Summary:
This research study assessed the relationship between alcohol use variables with measurements of multiple domains of heart rate variability (HRV), utilizing machine learning statistical analysis. Several other relevant factors known to influence HRV also were included in the models. Participants were young people categorized into three difference alcohol use categories, depending on their responses to the AUDIT. The authors present numerous results, many of which replicate known findings in the literature, however they are especially interesting given the Machine Analysis data analysis methodology modeled with data from a sample of healthy young people recruited from a primary health care setting. For example, although increased age is generally associated with worse HRV measures in the broader population, this study found that even among young people, heavy drinking was associated with worsened HRV measures.
Reviewer comments:
This paper is clearly written and reports on a study which was carefully designed and articulated. An additional strength is the comparison among participant alcohol consumption groups, recruited from young adults in a primary health care setting, which is a population often excluded from the research literature.
My suggested revisions are extremely minor:
- It may be beneficial to exercise caution when using the words “proven” and “proved” (an example is on Line 346). “Proof” implies a level of scientific certainty that is questionable to apply to this research area, which needs additional research to be fully understood. For example, an alternative option is to change the word on Line 478 from “proved” to “provided evidence”.
- Throughout the paper there often is a heavy use of language in favor of the Machine Learning statistical approach, however, it is important to note that there always are advantages and disadvantages to every statistical approach. Although the potential advantages of Machine Learning are many, results also can be misleading if the model is not specified correctly.
I suggest balancing the language regarding the enthusiasm with the Machine Learning approach to data analysis with language which shows understanding that Machine Learning is not without its problems. For example, consider changing the word “perfect” on line 418, to perhaps “optimum” or “useful”, etc.
- I was confused by the description “prospective study” on Line 480. My understanding is that a “prospective study” implies longitudinal measurement, which did not occur in the study described in this paper. This study was cross sectional in that participants were measured only at one time.
- Please define the abbreviation HF in the abstract (line 24). It also may be beneficial to define LF/HF as well. These abbreviations were not defined in the abstract in words before they were abbreviated; this may be confusing for the reader.
Author Response
Dear reviewer,
Thank you very much for your insightful comments on our paper. We appreciate the time and effort that you have dedicated to providing your valuable feedback on our manuscript. We have been able to incorporate the suggested changes and we have highlighted them within the manuscript with track changes.
This paper is clearly written and reports on a study which was carefully designed and articulated. An additional strength is the comparison among participant alcohol consumption groups, recruited from young adults in a primary health care setting, which is a population often excluded from the research literature.
My suggested revisions are extremely minor:
-
It may be beneficial to exercise caution when using the words “proven” and “proved” (an example is on Line 346). “Proof” implies a level of scientific certainty that is questionable to apply to this research area, which needs additional research to be fully understood. For example, an alternative option is to change the word on Line 478 from “proved” to “provided evidence”.
-
We changed that.
-
Throughout the paper there often is a heavy use of language in favor of the Machine Learning statistical approach, however, it is important to note that there always are advantages and disadvantages to every statistical approach. Although the potential advantages of Machine Learning are many, results also can be misleading if the model is not specified correctly.
I suggest balancing the language regarding the enthusiasm with the Machine Learning approach to data analysis with language which shows understanding that Machine Learning is not without its problems. For example, consider changing the word “perfect” on line 418, to perhaps “optimum” or “useful”, etc.
-
We changed that.
-
I was confused by the description “prospective study” on Line 480. My understanding is that a “prospective study” implies longitudinal measurement, which did not occur in the study described in this paper. This study was cross sectional in that participants were measured only at one time.
-
We corrected that.
-
Please define the abbreviation HF in the abstract (line 24). It also may be beneficial to define LF/HF as well. These abbreviations were not defined in the abstract in words before they were abbreviated; this may be confusing for the reader.
-
We defined the terms in the abstract.
Reviewer 2 Report
This is an excellent manuscript, addessing a topic of interest that has not been studied before.
Two minor remarks:
- Did the researchers write down any data concerning illicit drugs use that may affect the observed results?
- Do the authors assess any laboratory markers, such as troponin I, BNP or NT-proBNP at baseline?
Author Response
This is an excellent manuscript, addessing a topic of interest that has not been studied before.
Two minor remarks:
-
Did the researchers write down any data concerning illicit drugs use that may affect the observed results?
-
Do the authors assess any laboratory markers, such as troponin I, BNP or NT-proBNP at baseline?
Response
Dear reviewer,
Thank you very much for your insightful comments on our paper. We appreciate the time and effort that you have dedicated to providing your valuable feedback on our manuscript.
1. The participants who used drugs (opiates, cannabis, hallucinogenic and hypnotic substances, sedatives, cocaine, and other "stimulants") were excluded from the study. We have updated the manuscript accordingly (line 95).
2. The participant’s laboratory markers that we assessed were comprised of complete blood count and standard biochemistry (glycemia, cholesterol, GOT, GPT, sodium, potassium, creatinine, and uric acid). All the laboratory reports were normal and for this reason, we did not report them in the study. As the participants were healthy young adults, presenting no signs of acute or chronic heart failure, the standard laboratory markers for acute myocardial infarction or heart failure were not assessed.